# Assessment of the lifetime prevalence and incidence of induced abortion and correlates among female sex workers in Mombasa, Kenya: a secondary cohort analysis

Anne Marieke Simmelink ![ORCID],[1] Caroline M Gichuki,[1,2] Frances H Ampt,[3,4] Griffins Manguro,[2] Megan S C Lim,[3,4] Paul Agius,[3,4] Margaret Hellard,[3,4,5,6] Walter Jaoko,[7] Mark A Stoové,[3,4,8] Kelly L'Engle,[9] Marleen Temmerman,[2,10,11] Peter Gichangi,[2,11,12] Stanley Luchters[3,11,13,14]

For numbered affiliations see end of article.

**Correspondence to**
Stanley Luchters;
stanley.luchters@ceshhar.co.zw

## ABSTRACT

**Introduction** Prevalence of lifetime-induced abortion in female sex workers (FSWs) in Kenya was previously estimated between 43% and 86%. Our analysis aimed at assessing lifetime prevalence and correlates, and incidence and predictors of induced abortions among FSWs in Kenya.

**Methods** This is a secondary prospective cohort analysis using data collected as part of the WHISPER or SHOUT cluster-randomised trial in Mombasa, assessing effectiveness of an SMS-intervention to reduce incidence of unintended pregnancy. Eligible participants were current FSWs, 16–34 years and not pregnant or planning pregnancy. Baseline data on self-reported lifetime abortion, correlates and predictors were collected between September 2016 and May 2017. Abortion incidence was measured at 6-month and 12-month follow-up. A multivariable logistic regression model was used to assess correlates of lifetime abortion and discrete-time survival analysis was used to assess predictors of abortions during follow-up.

**Results** Among 866 eligible participants, lifetime abortion prevalence was 11.9%, while lifetime unintended pregnancy prevalence was 51.2%. Correlates of lifetime abortions were currently not using a highly effective contraceptive (adjusted OR (AOR)=1.76 (95% CI=1.11 to 2.79), p=0.017) and having ever-experienced intimate partner violence (IPV) (AOR=2.61 (95% CI=1.35 to 5.06), p=0.005). Incidence of unintended pregnancy and induced abortion were 15.5 and 3.9 per 100 women-years, respectively. No statistically significant associations were found between hazard of abortion and age, sex work duration, partner status, contraceptive use and IPV experience.

**Conclusion** Although experience of unintended pregnancy remains high, lifetime prevalence of abortion may have decreased among FSW in Kenya. Addressing IPV could further decrease induced abortions in this population.

**Trial registration number** ACTRN12616000852459.

## STRENGTHS AND LIMITATIONS OF THIS STUDY

⇒ This study presents incidence of abortion in a cluster-randomised cohort of female sex workers (FSWs).
⇒ It is the first to analyse predictors of abortions in FSWs, rather than correlates of past abortions only.
⇒ This paper explores a secondary research question, and the study was not originally powered to assess the predictors of abortions during follow-up.
⇒ The sensitive topic of abortions and sexual and reproductive health in general, might have resulted in a social desirability bias.

## INTRODUCTION

Research findings show that about 5% of Kenya's urban female reproductive population could be involved in sex work.[1] Female sex workers (FSWs) experience higher than average rates of HIV, other sexually transmitted infections (STIs) and unintended pregnancy.[2–5]

Unintended pregnancies often have negative consequences for FSWs, including financial adversity, social stigma and induced abortion.[4] In countries where abortion is illegal or difficult to access, women frequently resort to unsafe practices, risking severe medical complications.[6]

Reported prevalence of lifetime abortion among FSWs in low-income and middle-income countries (LMICs) varies from 24% in Laos in 2012 to 86% in 2000/2001 in Central and Western Kenya.[7–16] Many of these abortions are unsafe or sought in the informal sector.[8 11 16] In Mombasa, lifetime abortion among FSWs was estimated at 43% in 2008.[17]

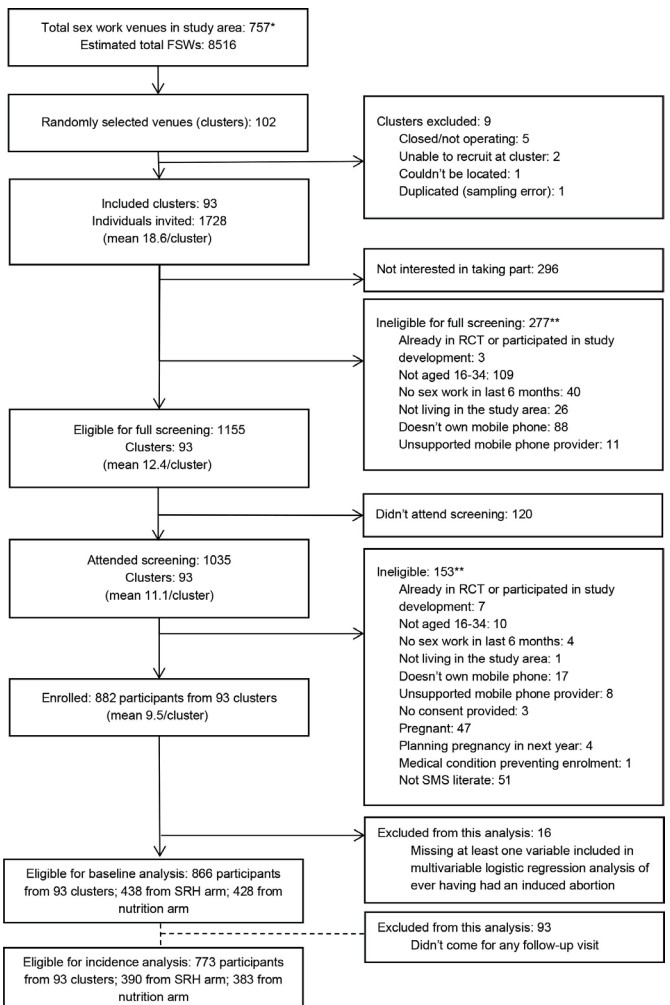

**Figure 1** Eligibility flow diagram for the WHISPER or SHOUT study as per CONSORT.[33 34] CONSORT, Consolidated Standards of Reporting Trials; FSW, female sex workers; RCT, randomised controlled trials; SRH, sexual and reproductive health.

In 2010, Kenya liberalised its abortion law, making abortion legal when 'there is need for emergency treatment, or the life or health of the mother is in danger'.[18] In practice, older laws criminalising abortion remain in place, creating ambiguity among health professionals with lawsuits remaining a threat.[19] Moreover, social, cultural and religious beliefs condemning pregnancy termination, misconceptions about the illegality and costs of the procedure, still hamper access to safe services for women.[4 19]

A national study in 2012 estimated an induced abortion rate of 48 per 1000 women among women aged 15–49, based on data of women who sought care for abortion complications.[20] It showed a diverse sociodemographic and economic background among these women in terms of educational level, employment status, marital status and religion.

Incidence of induced abortions among FSWs has not been studied in Kenya and studies analysing correlates of induced abortions elsewhere have been cross-sectional

and report correlates of lifetime abortions, precluding attribution of causality.[8–15 21] Examining incidence of abortions and identifying predictors will help inform future policies to improve care around abortions for FSWs and the need for longitudinal data about abortions in FSWs has therefore been recognised.[8 14]

This secondary data analysis aimed to examine lifetime prevalence and correlates, and incidence and predictors of induced abortions in a cluster-random sample of FSWs in Mombasa, Kenya.

## MATERIAL AND METHODS
### Study design
This is a secondary prospective cohort analysis using data collected as part of the WHISPER or SHOUT cluster-randomised trial in Mombasa. A detailed description of the study protocol can be found elsewhere.[22] In summary, the study was a two-arm cluster-randomised controlled trial assessing the effectiveness of two SMS-based interventions targeting sexual and reproductive health (SRH) and nutrition in FSWs in Mombasa, Kenya.

The study was conducted in two subcounties in Mombasa, Kisauni and Changamwe, between September 2016 and July 2018. Ninety-three venues were randomly sampled with a probability proportional to FSW population size at the venue (figure 1). Trained community mobilisers and peer educators recruited FSWs from the venues until the required sample size of 860 FSWs was achieved. This study uses baseline and follow-up data from the trial. During the recruitment period of 14 September 2016 to 16 May 2017, 882 women were enrolled in the study. Follow-up continued until 31 July 2018. A subsample of 866 women (98.2%) was analysed for this secondary analysis (figure 1).

### Study participants and procedures
Women were eligible for the study if they were aged 16–34 years; self-reported to have engaged in paid sex work in the last 6 months; were reportedly not pregnant or planning pregnancy within the next 12 months; resided within the study area and were able to read text messages in basic English. Study-specific community mobilisers visited the selected clusters to recruit FSWs and conducted prescreening interviews identifying women who self-reported to be sex workers. Potentially eligible FSWs were referred to the nearest study clinic for a clinical assessment, including a urine pregnancy test, and STI and HIV testing. Enrolled participants then completed a structured questionnaire administered in Swahili by trained research assistants, who had previously participated in research with FSWs. The questionnaire captured detailed sociodemographic, SRH information. Follow-up visits were scheduled at 6 and 12 months after enrolment. Procedures at follow-up visits were similar to those done at enrolment.

Participants received two to three SMS per week for 12 months. The messages consisted of stand-alone push

messages, role model stories and on-demand messages, accessed using assigned codes. Participants only received and accessed messages on their phones from their assigned intervention.

## Outcomes and correlates

Lifetime prevalence of induced abortion was assessed at baseline with the question 'How many times have you ever had an induced abortion?'. Induced abortions were assessed during follow-up, by asking participants if they had been pregnant since their last visit. The outcome of each reported pregnancy was then assessed, and in the case of an induced abortion, the location was documented. Formal sector abortions were those taking place at a government or private hospital/clinic, a private doctor/general practitioner or a family planning (FP) clinic. Informal sector abortions were defined as those taking place at home, a pharmacy or traditional healer.[23]

Pregnancies during follow-up were confirmed with a urine pregnancy test at the study clinic, or self-reported by the participant when occurring between study visits. Pregnancy intention for all reported pregnancies was assessed using the London Measure of Unintended Pregnancy (LMUP), a six-item scale. A pregnancy scoring less than 10 out of 12 on the LMUP was defined as unintended.[24]

All correlates of lifetime-induced abortions and predictors of incident induced abortions were self-reported at baseline. Use of a highly effective contraceptive method was defined as the use of contraceptive implants, intrauterine devices (IUD), injection, contraceptive pill and sterilisation. High knowledge on FP was defined as answering five out of six true–false statements on FP correctly. Having a positive attitude on FP was defined as agreeing with at least three out of four positive attitude statements. Self-efficacy in FP was defined as high when agreeing with two statements on this topic. Household socioeconomic status tertiles were generated using principal component analysis, based on 12 household assets.[25] Prevalence of intimate partner violence (IPV) was assessed by asking if the participant had ever been pushed, slapped, hit or kicked by a partner or had been physically forced to have sex, agreed to sex out of fear of the consequences or forced to do something sexual that she perceived as degrading or humiliating. For each item, it was also assessed if this happened in the previous 12 months.

## Statistical analysis

Statistical analysis was performed using Stata software V.14 (StataCorp, College Station, TX, USA). Distributions of sociodemographic characteristics, reproductive history, contraceptive use and IPV at baseline were explored with means and SD for continuous variables and proportions for categorical variables. Covariates age, education level, SES-tertile, duration of sex work, having a husband/boyfriend, use of highly effective contraceptives and experience of IPV were included in the multivariate models on the basis of a review of literature.[8–15 21] Covariates high FP knowledge, positive attitude to FP and high FP specific self-efficacy were included in the multivariate models on the basis of theoretical assumptions by the coauthors. It was hypothesised that these characteristics would be positively associated with contraceptive use and would protect against experiencing an induced abortion. Correlates of lifetime abortion were identified using weighted multivariable logistic regression. Associations were considered statistically significant at the 5% level. The outcome incident abortion was interval-censored (measured at 6-monthly intervals). Therefore, a discrete-time survival analysis was performed using generalised linear mixed (GLM) modelling with complementary log–log link function and binomial distribution, a method that produces estimated HRs. Abortions during follow-up were analysed for all participants who attended at least one follow-up visit. All outcomes presented here are cluster-adjusted, based on inverse probability sample means to account for sampling bias.

## Patient and public involvement

Patients or the public were not directly involved in the design, or conduct, or reporting, or dissemination plans of our research.

## RESULTS

Mean age was 25.5 years (SD=4.7) (table 1). The majority of women (n=765, 88.7%) had at least completed primary education and 306 (34.9%) had completed secondary education. Just over half of the participants (n=484, 56.6%) reported to have a current husband or boyfriend, but 812 (94.0%) reported not to live with a partner. Mean duration of employment in sex work was 4.7 years (SD=3.5). Among women currently reporting a husband or boyfriend, 344/483 (70.8%) had not disclosed their employment in sex work to their partners; 605/861 women (68.9%) worked fulltime in sex work and 508/861 (59.8%) earned more than 2000 Ksh (about US$20) per week from sex work. The majority of women (n=666, 76.1%) had ever been pregnant, and 451/864 (51.2%) ever had an unintended pregnancy; 103/866 (11.9%) reported to have had at least one induced abortion in their lifetime. Among women who had an induced abortion, 58/102 (57.1%) went to a private hospital or clinic for the most recent abortion; 29/102 (29.1%) women had their most recent abortion in the informal sector, like home, a pharmacy or traditional healer. At baseline, 473 women (54.4%) reported to use a highly effective contraceptive method. Three-quarters of FSW (650/866; 75.0%) ever experienced IPV and 525/866 (60.1%) experienced IPV in the past 12 months the before the baseline questionnaire was conducted.

Women currently not using a highly effective contraceptive (adjusted OR (AOR)=1.76 (95% CI=1.11 to 2.79), p=0.017) and women who ever experienced IPV (AOR=2.61 (95% CI=1.35 to 5.06), p=0.005) were significantly more likely to report a history of induced abortion,

**Table 1** Sociodemographic characteristics, reproductive history, contraceptive use and intimate partner violence at baseline of female sex workers in Mombasa, Kenya (n=866, unless stated otherwise)

| Characteristic | N | Cluster-adjusted mean (SD) or proportion in % (95% CI)* |
|---|---|---|
| Mean age, in years | | 25.5 (4.7) |
| Highest level of education | | |
| None or some primary | 101 | 11.2 (9.2 to 13.6) |
| Completed primary or some secondary | 459 | 53.8 (50.1 to 57.6) |
| Completed secondary or some tertiary | 306 | 34.9 (31.3 to 38.7) |
| Religion (n=864) | | |
| Protestant | 389 | 44.8 (41.4 to 48.2) |
| Catholic | 304 | 36.0 (32.3 to 39.8) |
| Muslim | 171 | 19.2 (15.6 to 23.4) |
| Electricity available in household (n=863)† | 660 | 76.4 (73.0 to 79.5) |
| Duration of sex work, in years | | 4.7 (3.5) |
| Full time FSW (n=861)‡ | 605 | 68.9 (64.0 to 73.5) |
| Weekly income from sex work (n=861) | | |
| ≤1000 Ksh | 144 | 16.2 (13.2 to 19.8) |
| 1001–2000 Ksh | 211 | 24.0 (21.0 to 27.3) |
| ≥2001 Ksh§ | 508 | 59.8 (55.1 to 64.3) |
| Sex work venue | | |
| Bar with lodging | 388 | 43.8 (37.7 to 50.2) |
| Bar without lodging | 147 | 17.2 (13.7 to 21.3) |
| Lodging/guesthouse | 138 | 15.1 (10.9 to 20.5) |
| Street/beach | 86 | 11.2 (7.8 to 15.7) |
| Other¶ | 107 | 12.7 (9.1 to 17.5) |
| Marital status | | |
| Married/cohabiting | 54 | 6.0 (4.5 to 8.0) |
| Single (not cohabiting) | 627 | 73.0 (69.6 to 76.2) |
| Separated/divorced/widowed | 185 | 21.0 (18.0 to 24.3) |
| Currently has husband/boyfriend | 484 | 56.6 (52.5 to 60.7) |
| Disclosure of sex work to husband/boyfriend (n=483)** | | |
| Yes | 136 | 28.7 (24.4 to 33.3) |
| No | 344 | 70.8 (66.0 to 75.1) |
| Do not know | 3 | 0.6 (0.2 to 1.7) |
| Ever had a pregnancy | 666 | 76.1 (72.3 to 79.5) |
| Has a living child | 622 | 71.2 (67.1 to 75.0) |
| Ever had an unintended pregnancy (n=864)†† | 451 | 51.2 (47.4 to 54.9) |
| Ever had an induced abortion | 103 | 11.9 (10.0 to 14.2) |
| Location of most recent induced abortion (n=102) | | |
| Government hospital | 2 | 1.8 (0.4 to 6.9) |
| FP clinic, like Marie Stopes | 9 | 8.3 (4.4 to 15.0) |
| Private hospital/clinic | 58 | 57.1 (46.5 to 67.2) |
| Private doctor GP | 4 | 3.7 (1.4 to 9.6) |
| Pharmacy | 9 | 8.9 (4.6 to 16.8) |

Continued

**Table 1** Continued

| Characteristic | N | Cluster-adjusted mean (SD) or proportion in % (95% CI)* |
|---|---|---|
| Traditional healer | 4 | 3.8 (1.4 to 10.0) |
| Home | 16 | 16.4 (9.0 to 27.8) |
| Uses a highly effective contraceptive method‡‡ | 473 | 54.4 (49.5 to 59.2) |
| Ever experienced IPV | 650 | 75.0 (71.1 to 78.5) |
| Experienced IPV in past 12 months§§ | 525 | 60.1 (55.5 to 64.6) |

*Inverse probability-weighted percentages.
†Availability of electricity in the household is presented here as a proxy for household SES.
‡Fulltime work as FSW is characterised as having no other sources of income in the last 6 months.
§1000 Kenyan Shilling (Ksh) is about US$10.
¶Brothel, strip club, casino, massage parlors, parks or home.
**Among participants with a husband or boyfriend.
††Assessed using the London Measure of Unintended Pregnancy (LMUP).
‡‡Highly effective is defined as use of contraceptive implants, IUD, injection, oral contraceptive pill and sterilisation.
§§Before baseline.
FP, family planning; FSW, female sex workers; GP, general practitioner; IPV, intimate partner violence; IUD, intrauterine devices; SES, socioeconomic status.

when controlled for potential confounders (table 2). Longer duration of sex work showed a borderline positive association with history of abortion (AOR=1.08 (95% CI=1.00 to 1.16), p=0.053). Although higher age was significantly associated with a history of abortion in bivariate analysis, after adjusting for confounding factors, this association was no longer seen.

During the study follow-up, 773 women attended at least one follow-up visit (figure 1). Total follow-up time was 9468 months, with an average of 12.2 months per woman (data not shown). A total of 131 participants became pregnant, with a total of 145 pregnancies among these women (figure 2). Of these pregnancies, 122/145 were unintended according to the LMUP. 31 out of 145 pregnancies ended in induced abortion, among 29 women and across 789 women-years at risk. Overall incidence rate was 3.9 induced abortions per 100 women-years of observation. Out of 31 abortions, 19 took place in the formal sector and 12 in an informal setting.

The GLM modelling of abortion incidence showed that women experiencing IPV in the past year (HR=1.93 (95% CI=0.86 to 4.34), p=0.122) and women not using a highly effective contraceptive (HR=1.51 (95% CI=0.66 to 3.49), p=0.332) exhibited a higher hazard of abortion, independent of other factors, although these results were not significant (table 3). We did not find a relation between age, mean duration of sex work, currently having a husband or boyfriend and the intervention under study and hazard of induced abortion.

**Table 2** Correlates of participants with a history of induced abortion, and cluster-adjusted bivariable and multivariable logistic regression analysis on history of induced abortion (n=866)

| Characteristic | Ever had an induced abortion (n=103); n/N (cluster-adjusted proportion in %)* | Crude OR | | Adjusted OR | |
|---|---|---|---|---|---|
| | | OR (95% CI)† | P value | OR (95% CI)† | P value |
| Age (in years) | 27.0 (4.9)‡ | 1.08 (1.03 to 1.14) | 0.001 | 1.04 (0.97 to 1.11) | 0.28 |
| Highest level of education | | | | | |
| None or some primary | 14/101 (13.5) | Ref. | | Ref. | |
| Completed primary or some secondary | 57/459 (12.4) | 0.90 (0.47 to 1.72) | 0.749 | 0.96 (0.48 to 1.99) | 0.895 |
| Completed secondary or some tertiary | 32/306 (10.7) | 0.77 (0.40 to 1.47) | 0.423 | 0.83 (0.40 to 1.84) | 0.62 |
| SES-tertile | | | | | |
| Poorest | 39/290 (13.4) | Ref. | | Ref. | |
| Middle | 33/287 (10.9) | 0.79 (0.46 to 1.35) | 0.386 | 0.89 (0.50 to 1.59) | 0.697 |
| Richest | 31/289 (11.4) | 0.83 (0.47 to 1.46) | 0.51 | 0.83 (0.46 to 1.50) | 0.674 |
| Mean duration of sex work (in years) | 6.1 (3.4)‡ | 1.12 (1.07 to 1.17) | <0.001 | 1.08 (1.00 to 1.16) | 0.053 |
| Highly effective contraceptive use§ | | | | | |
| Yes | 51/473 (10.6) | Ref. | | Ref. | |
| No | 52/393 (13.5) | 1.32 (0.88 to 1.97) | 0.173 | 1.76 (1.11 to 2.79) | 0.017 |
| High FP knowledge score | | | | | |
| No | 60/562 (10.5) | Ref. | | Ref. | |
| Yes | 43/304 (14.5) | 1.44 (0.95 to 2.19) | 0.084 | 1.34 (0.85 to 2.10) | 0.2 |
| Positive attitude to FP use | | | | | |
| No | 43/354 (12.4) | Ref. | | Ref. | |
| Yes | 60/512 (11.6) | 0.93 (0.60 to 1.44) | 0.743 | 0.90 (0.56 to 1.45) | 0.661 |
| High FP-specific self-efficacy | | | | | |
| No | 24/237 (10.1) | Ref. | | Ref. | |
| Yes | 79/629 (12.6) | 1.28 (0.76 to 2.17) | 0.345 | 1.23 (0.72 to 2.10) | 0.454 |
| Ever-experienced intimate partner violence | | | | | |
| No | 12/216 (5.2) | Ref. | | Ref. | |
| Yes | 91/650 (14.1) | 2.98 (1.55 to 5.74) | 0.001 | 2.61 (1.35 to 5.06) | 0.005 |

*Inverse probability-weighted percentages.
†SEs are corrected by cluster sandwich variance estimation.
‡Mean (SD) of women who ever had an induced abortion.
§Highly effective is defined as use of contraceptive implants, IUD, injection, contraceptive pill and sterilisation.
FP, family planning; IUD, intrauterine devices; SES, socioeconomic status.

## Discussion

This study adds to the current knowledge of abortion practices in FSWs. Lifetime-induced abortion prevalence in this population was 11.9%. This seems considerably lower than previous figures of lifetime abortion of 86% in 2004 in central and western Kenya among FSWs of a similar age, and 43% in 2008 in Mombasa among FSWs who were on average 2 years older.[7 17] In the former study, it was not specified if these abortions also included spontaneous abortions, which might have overestimated the prevalence of abortions. However, despite the sociodemographic and methodological differences between the studies, the size of the difference is suggestive of an actual lower rate in abortions in this population. The prevalence of abortions found here is also lower than reports from other LMICs ranging between 24% and 64% in Laos and

Cote d'Ivoire, respectively.[8–16] A possible explanation for this lower prevalence is the relatively high use of highly effective contraceptives of 54% in our cohort, compared with similar studies from LMICs.[8 12 14–16] The findings are furthermore consistent with a lower-than-expected HIV prevalence and unintended pregnancy incidence in our cohort and could be a result of peer-mediated interventions implemented over the past years in the Mombasa area.[26 27] These have mostly targeted prevention of HIV and STIs, but likely have had a lowering effect on unintended pregnancies and induced abortions as well.[22 26] Furthermore, this study, in contrast to above referenced studies, attempted to draw a representative sample of an FSW population from community settings, whereas other studies used non-probability sampling methods, which might have overestimated past abortions.

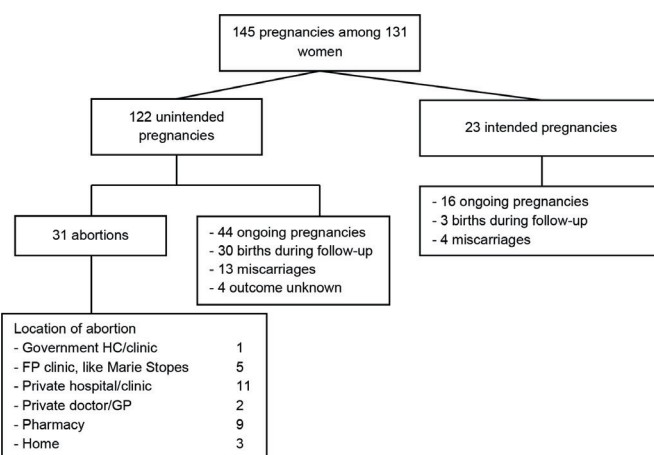

**Figure 2** Overview of pregnancy outcomes during the 12-month follow-up. n=773. FP, family planning; GP, general practitioner.

Despite the lower-than-expected unintended pregnancy incidence, still 51% of FSWs in our cohort reported an unintended pregnancy in their lifetime. The gap between lifetime unintended pregnancies and lifetime-induced abortions could indicate that many women decide to keep a child from unintended pregnancies, which could be supported by the fact that between 70% and 80% of young FSW in Mombasa have reported one or more children.[4 27] It may also indicate a high unmet need for induced abortion services among FSWs, for example due to ongoing or increasing difficulties in accessing SRH or abortion services for this group or increasing sociocultural barriers to abortion. Barriers to accessing other SRH services such as long-acting reversible contraceptives has previously been reported for this population.[28]

The present study is one of the few studies to report incidence of abortion among FSWs and to the best of our knowledge, the first to analyse predictors of abortions in FSWs, rather than correlates of past abortions only. Incidence of induced abortion in our cohort was 3.9 per 100 women-years. Compared with other studies from LMICs, this is similar to two studies reporting abortion incidence rates of 3.1 and 3.0 per 100 women-years and lower than a third study reporting 7.4 induced abortions per 100 women-years among FSWs.[29] The intervention under study had no measurable effect on unintended pregnancy incidence and is therefore unlikely to have affected incidence of induced abortions.[27]

Informal sector abortions where common in this cohort, with 29% of women having had their most recent abortion in the informal sector, and 39% of the reported abortions during follow-up happening in the informal sector. These informal sector abortions, put women at higher risk of complications due to unsafe practices and this denotes a need for information on safer alternatives, like the Marie Stopes clinics.[23]

Multiple studies have found both age and duration of sex work to be correlated to past abortions. Commonly higher age[8 11 15] and longer duration[9 12] of sex work were associated with higher lifetime abortion prevalence. One study found that younger age was associated with past abortions.[10] In our cohort, although FSWs with a past abortion in our cohort were older in the crude analysis, after adjusting for other correlates, this difference was no longer significant. The association with longer duration of sex work remained borderline significant in multivariate analysis. We did not find a relation between age and mean duration of sex work and having an induced abortion during follow-up. This might suggest that the association between past abortions and higher age and longer duration of sex work is caused by cumulative exposure to high risk of pregnancies and abortion.

We found a positive association between currently not using a highly effective contraceptive and having a past abortion. No difference was found in FP-specific self-efficacy or knowledge, or attitude towards FP among women with and without a past abortion. The found association could indicate significant barriers to uptake or

**Table 3** Baseline predictors of incident abortion in FSWs among Mombasa, Kenya (n=773)*

| Baseline predictors of incident abortions | Unadjusted | | Adjusted HR† | |
|---|---|---|---|---|
| | HR (95% CI)‡ | P value | HR (95% CI)‡ | P value |
| Age (in years) | 0.96 (0.89 to 1.04) | 0.315 | 1.00 (0.91 to 1.09) | 0.918 |
| Mean duration of sex work (in years) | 0.93 (0.82 to 1.05) | 0.234 | 0.92 (0.79 to 1.09) | 0.336 |
| Currently has husband/boyfriend | 0.83 (0.40 to 1.73) | 0.622 | 0.80 (0.39 to 1.64) | 0.537 |
| Not using highly effective contraceptive§ | 1.50 (0.69 to 3.23) | 0.310 | 1.51 (0.66 to 3.49) | 0.332 |
| Experienced IPV in last 12 months | 1.67 (0.74 to 3.79) | 0.216 | 1.93 (0.86 to 4.34) | 0.122 |

*Discrete-time survival analysis including the first induced abortion per women. Generalised linear mixed model with complementary log–log link, binomial distribution, offset for log time between visits and random intercept for participants.
†All adjusted HRs are also adjusted for the intervention. The intervention had no detectable effect on the outcome of incident abortions.
‡Cluster robust SE s for sex-work venue clustering.
§Highly effective is defined as use of contraceptive implants, IUD, injection, oral contraceptive pill and sterilisation.
FSW, female sex workers; IPV, intimate partner violence; IUD, intrauterine devices.

continuation of a highly effective contraceptive method post abortion, as has previously been acknowledged by a study in Kenya.[30]

In our cohort, experience of IPV was high and the odds of having had a past abortion were more than 2.5-times as high for women who experienced IPV in the past, consistent with findings from other studies.[14 15] Our study also showed a positive association between experience of IPV in the past 12 months and abortions during follow-up, but this was not significant. Experience of (intimate partner) violence has been shown to have a negative effect on the reproductive health of FSWs, with greater risks of adverse pregnancy outcomes and forced termination of pregnancy.[31 32] Addressing the problem of IPV in this population could further lower induced abortions.

## Limitations

Some limitations should be considered when interpreting the findings from this study. The sensitive topic of abortions and SRH in general, might have resulted in a social desirability bias. To minimise this, peer-educators and research assistants had previous experience working with the target population and received additional training. Attrition bias might have occurred due to loss to follow-up of pregnant participants, as has been recognised by anecdotal evidence.[27] This might have resulted in an underestimation of abortions in our study. The robust multistage sampling method improved the ability to generalise the findings to the larger sex work population in the Coast region. However, this study was done in a well-researched population, targeted by other peer-mediated interventions in the past two decades, which may limit generalisation to sex worker populations in other settings. A further limitation is that this paper explores a secondary research question, and the study was not originally powered to assess the predictors of abortions during follow-up. Unknown timing of the past abortions in relation to studied correlates, restrict judgement of temporality of the studied associations. Finally, measurements of abortions stopped when the intervention stopped, so the actual number of abortions during follow-up might in fact be higher than captured in the study.

## Suggestions for further research

Future research is needed to explore the trend in abortion incidence among FSWs in Kenya. In order to improve care, we need to better understand current abortion practices, the decision-making process around terminating unintended pregnancies, how uptake of highly effective contraceptives can be increased postabortion, as well as the relationship between experience of IPV and induced abortions.

## Conclusions

In conclusion, the prevalence of lifetime-induced abortions in a random cohort of FSWs in Mombasa was 11.9% and incidence was 3.9 per 100 women-years, whereas prevalence and incidence of unintended pregnancies were higher at 51% and 15.5 per 100 women-years, respectively. A history of induced abortion was positively associated with not using a highly contraceptive method at baseline and having experienced IPV in the past. The study did not find a significant association with the studied predictors of abortions.

**Author affiliations**
[1]Department of Population Health, The Aga Khan University, Nairobi, Nairobi, Kenya
[2]International Centre for Reproductive Health Kenya, Mombasa, Kenya
[3]Burnet Institute, Melbourne, Victoria, Australia
[4]Department of Epidemiology and Preventive Medicine, Monash University, Clayton, Victoria, Australia
[5]Department of Infectious Diseases, The Alfred Hospital, Melbourne, Victoria, Australia
[6]Doherty Institute and School of Population and Global Health, University of Melbourne, Melbourne, Victoria, Australia
[7]Department of Medical Microbiology and Immunology, University of Nairobi, Nairobi, Nairobi, Kenya
[8]School of Psychology and Public Health, La Trobe University, Melbourne, Victoria, Australia
[9]School of Nursing and Health Professions, University of San Francisco, San Francisco, California, USA
[10]Department of Obstetrics and Gynaecology, The Aga Khan University Hospital Nairobi, Nairobi, Kenya
[11]Department of Public Health and Primary Care, Ghent University, Gent, Belgium
[12]Technical University of Mombasa, Mombasa, Kenya
[13]Centre for Sexual Health and HIV/AIDS Research (CeSHHAR), Harare, Zimbabwe
[14]Liverpool School of Tropical Medicine (LSTM), Liverpool, UK

**Acknowledgements** The authors would like to acknowledge the hard work of the field research team, specifically Christine Maghanga, Millicent Okello, Betty Kitili, Judith Wamaua, Elizabeth Bilasi, Marion Mwangi, Promillah Muindi, Rukia Abdallah and Khadija Kassim, as well as all the study participants of the study. The authors gratefully acknowledge the contribution of funding from Australia's National Health and Medical Research Council for the WHISPER or SHOUT trial (Project Grant GNT 1087006), Career Development Fellowships for SL, Senior Research Fellowship for MS and a Postgraduate Scholarship for FHA. The sponsor did not contribute to study design; data collection, analysis or interpretation; manuscript writing or the decision to submit the article for publication. Finally, we acknowledge the contribution of funding from the Victorian Operational Infrastructure Support Programme received by the Burnet Institute.

**Contributors** SL was the principal investigator on the study and is the guarantor of this work. SL, FHA, PG, MSCL, PA, MH, WJ, MAS, KLE, and MT contributed to the study design. CMG and GM coordinated the trial and undertook data acquisition in Kenya under the supervision of PG. AMS and SL conceptualised the manuscript. AMS and PAA conducted the statistical analyses. AMS and CG wrote the first draft of the manuscript. All authors contributed to data interpretation, provided critical input and approved the final version of the manuscript.

**Funding** This work was supported by Australia's National Health and Medical Research Council (NHMRC), Project Grant GNT 1087006.

**Competing interests** None declared.

**Patient and public involvement** Patients and/or the public were not involved in the design, or conduct, or reporting or dissemination plans of this research.

**Patient consent for publication** Consent obtained directly from patient(s).

**Ethics approval** This study involves human participants. The study was approved by the Kenyatta National Hospital and University of Nairobi Ethics and Research Committee, Kenya (KNH-UoN ERC-KNH-ERC/RR/493) and the Monash University Human Research Ethics Committee, Australia (MUHREC-CF16/1552-2016000812). Participants gave informed consent to participate in the study before taking part.

**Provenance and peer review** Not commissioned; externally peer reviewed.

**Data availability statement** All data relevant to the study are included in the article or uploaded as supplementary information. Raw data are available upon reasonable request. The original data are not available in a public repository. The corresponding author is to be contacted for the consideration of any data requests.

**ORCID iD**
Anne Marieke Simmelink http://orcid.org/0000-0001-5245-8388

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
