## [Reviewer comments · BMJ Open]

ARTICLE DETAILS

TITLE (PROVISIONAL)	ASSESSMENT OF THE LIFETIME PREVALENCE AND INCIDENCE OF INDUCED ABORTION AND CORRELATES AMONG FEMALE SEX WORKERS IN MOMBASA, KENYA: A SECONDARY COHORT ANALYSIS
AUTHORS	Simmelink, Anne Marieke; Gichuki, Caroline M.; Ampt, Frances H.; Manguro, Griffins; Lim, Megan; Agius, Paul; Hellard, Margaret; Jaoko, Walter; Stoové, Mark; L'Engle, Kelly; Temmerman, Marleen; Gichangi, Peter; Luchters, Stanley

VERSION 1 – REVIEW

REVIEWER	Awungafac, George Karolinska Institute, Global Public Health
REVIEW RETURNED	18-Sep-2021

GENERAL COMMENTS	Abstract Line 34-35: I suggest “our secondary objective aimed to assess” be changed to “our study aimed at assessing”. Line 47: The aspect “with available data on outcome and exposure variables” is not relevant in the abstract. Authors should consider deleting this part and report it in the text. Introduction General comment; Authors have stated the need for longitudinal data on abortion in FSWs. But they have not provided adequate explanation to expose the research gap concerning this topic in Kenya. The introduction is only concentrated on lifetime abortion prevalence which is only a part of the study. To better place the study in context, it will also be useful for to include a brief overview of sociodemographic factors (literacy rates, employment, fertility rates, etc.) that determine the sexual and reproductive health of women in Kenya. Materials and methods Study design The authors have focused this section more on presenting how the WHISPER or SHOUT trial was conducted and the sampling of FSWs for the SMS intervention trial. This is fine as background information about the origin of this study. An experimental design itself cannot, for instance, be set up to estimate lifetime prevalence or incidence of abortion. Even though they are analyzing data collected during the trial, they should be reporting an observational study as the topic indicates. If baseline data (for example) was utilized, then the data probably came from a cross-sectional baseline assessment during the trial. Authors should further describe the observational study designs supporting the analysis used in this study.
---

	Line 112-113: It is not clear why authors have explained how they did randomization whereas they have referred readers to the study protocol. This sentence does not seem relevant to include in this study which clearly is observational. Line 121: I Suggest authors rephrase “were not pregnant” to “were reportedly not pregnant” since they are still conducting objective pregnancy testing (Line 125) to confirm eligibility. Line 124: Authors should explain the purpose of the pre-screening interviews? Line 131-134: It is not clear why authors included the trial interventions in this section. They are detailing the procedures of the intervention when, in fact, the current analysis is not findings of a cluster trial. This study is clearly an observational study and should be reported as such even if the data was collected during a trial. Line 136: The sub-heading “Outcomes” should be changed to a more appropriate title. Authors are describing measurements of the study variables. The main outcomes under study being “lifetime prevalence and incidence of induced abortion”, and then the correlates. So, all variables cannot be called outcomes. Line 142: GP and FP should first be written in full. Line 156: Authors should include the reference for principal component analysis. line 162-173: It is logical to begin this section with the descriptive statistics before moving on to describe the advanced statistical approaches used. Line 164-165: Authors state that “covariates for multivariate models were determined on the basis of a review of literature or theoretical assumptions by the co-authors”. Meanwhile this may be clear for some covariates, readers may not understand this reasoning for many covariates; for example, experiencing IPV in the past 12 months. Authors should briefly provide the findings and/or thoughts supporting their selection of covariates for the multivariate models other than the sociodemographic characteristics, especially experiencing IPV in the preceding 12 months. Ethics Even though author indicated having obtained ethical approval, the ethical procedures applied in this study are not written in the text. Ethical considerations should be included in the manuscript. Results Lines 179-181: I Suggest this section should be moved to the relevant section under “materials and methods”. Discussion The researchers have attributed the “actual reduction in abortions in this population (line 246)” to unmet needs for induced abortion among FSWs, due to difficulty in accessing abortion services. Could they discuss any ongoing interventions in Kenya/Mombasa targeting FSWs for SRH or any reported programmatic challenges in providing SRH in this population. Could they also discuss any other potential motivations that can trigger an intention to abort or keep a pregnancy in FSWs? Line 273-274: Authors state that “...30% of women having their most recent abortions in the informal sector”. In table 1 of the main text, 29 out of 102 FSWs had their most recent induced abortion in the informal sector (pharmacy, traditional healer and home), which corresponds to 28% and not 30% as reported. Conclusions
--	---

	line 329-330: I suggest authors should avoid using the phrase; “this was to our knowledge the first study attempting to identify predictors of abortion in FSWs” in the conclusion. Supplemental information Table 1 & 2: The STROBE checklist would have rather been used to report this study.
--	--

VERSION 1 – AUTHOR RESPONSE

Reviewer: 1

Dr. George Awungafac, Karolinska Institute

Abstract

- Line 34-35: I suggest “our secondary objective aimed to assess” be changed to “our study aimed at assessing”.

Response: As suggested, we have changed the sentence, and it now reads:” Our analysis aimed at assessing lifetime prevalence and correlates, and incidence and predictors of induced abortions among FSWs in Kenya”.

- Line 47: The aspect “with available data on outcome and exposure variables” is not relevant in the abstract. Authors should consider deleting this part and report it in the text.

Response: As suggested, we have removed this from the abstract.

Introduction

- Authors have stated the need for longitudinal data on abortion in FSWs. But they have not provided adequate explanation to expose the research gap concerning this topic in Kenya. The introduction is only concentrated on lifetime abortion prevalence which is only a part of the study. To better place the study in context, it will also be useful for to include a brief overview of sociodemographic factors (literacy rates, employment, fertility rates, etc.) that determine the sexual and reproductive health of women in Kenya.

Response: We appreciate the feedback and have now elaborated further on the research gap in Kenya and on the available data about incidence of abortion in Kenya: ‘A national study estimated an induced abortion rate of 48 per 1,000 women among women aged 15-49 in Kenya in 2012, based on data of women who sought care for abortion complications.[20] [...]

Incidence of induced abortions among FSWs has not been studied in Kenya and studies analysing correlates of induced abortions elsewhere have been cross-sectional and report correlates of lifetime abortions, precluding attribution of causality.[8–15,21] Examining incidence of abortions and identifying predictors will help inform future policies to improve care around abortions for FSWs and the need for longitudinal data about abortions in FSWs has therefore been recognised.[8,14]’ In addition, some information has been given about the sociodemographic and economic background of women who seek abortion care (which has found to be heterogeneous), though we argue that in this short introduction it is difficult to sketch a complete profile of these women, it now reads: ‘[...] It showed a diverse sociodemographic and economic background among these women in terms of educational level, employment status, marital status and religion.’

Materials and methods

- The authors have focused this section more on presenting how the WHISPER or SHOUT trial was conducted and the sampling of FSWs for the SMS intervention trial. This is fine as background

information about the origin of this study. An experimental design itself cannot, for instance, be set up to estimate lifetime prevalence or incidence of abortion. Even though they are analyzing data collected during the trial, they should be reporting an observational study as the topic indicates. If baseline data (for example) was utilized, then the data probably came from a cross-sectional baseline assessment during the trial. Authors should further describe the observational study designs supporting the analysis used in this study.

Response: We noted the challenges our description of the study design gave to both the Editor and the Reviewer. We have now more clearly stated that this is a secondary analysis, and applied a prospective cohort design, using the data from a cluster-randomized trial. The description of the sampling (i.e. a cluster-random sample) remains critically important for the description of the included population.

- Line 112-113: It is not clear why authors have explained how they did randomization whereas they have referred readers to the study protocol. This sentence does not seem relevant to include in this study which clearly is observational.

Response: As suggested by the reviewer, we have deleted the sentence stating: 'Sex-work venues were randomized to either the SRH or nutrition intervention group after the cluster was fully enrolled and baseline data obtained.'

- Line 121: I Suggest authors rephrase "were not pregnant" to "were reportedly not pregnant" since they are still conducting objective pregnancy testing (Line 125) to confirm eligibility.

Response: This has been changed.

- Line 124: Authors should explain the purpose of the pre-screening interviews?

Response: Community mobilizers went out to the recruitment sites for pre-screening, which aimed at identifying women who self-reported to be sex workers. Potentially eligible women were referred to the study clinic where the full eligibility criteria were applied. This has now been clarified in the text, and it reads: "Study-specific community mobilizers visited the selected clusters to recruit FSWs and conducted pre-screening interviews identifying women who self-reported to be sex workers. Potentially eligible FSWs were referred to the nearest study clinic.....".

- Line 131-134: It is not clear why authors included the trial interventions in this section. They are detailing the procedures of the intervention when, in fact, the current analysis is not findings of a cluster trial. This study is clearly an observational study and should be reported as such even if the data was collected during a trial.

Response: We understand the feedback from the reviewer as the description of the intervention is not a critical part of this analysis. However, in our analysis and also briefly described in the results, we did check if the intervention (SMS messages) had any influence on the hazard of induced abortion. We would therefore argue it may be worthwhile leaving the very brief description of the intervention for easier understanding of the readers.

- Line 136: The sub-heading "Outcomes" should be changed to a more appropriate title. Authors are describing measurements of the study variables. The main outcomes under study being "lifetime prevalence and incidence of induced abortion", and then the correlates. So, all variables cannot be called outcomes.

Response: Thanks for the feedback and we have changed the heading to: 'Outcomes and correlates'.

- Line 142: GP and FP should first be written in full.

Response: This has been changed.

- Line 156: Authors should include the reference for principal component analysis.

Response: we added a reference (Vyas 2006) for the principal component analysis, as requested.

- Line 162-173: It is logical to begin this section with the descriptive statistics before moving on to describe the advanced statistical approaches used.

Response: We appreciate the feedback and have added a sentence describing the descriptive analysis. The section now reads: "Distributions of socio-demographic characteristics, reproductive history, contraceptive use and intimate partner violence at baseline were explored with means and standard deviations for continuous variables and proportions for categorical variables."

- Line 164-165: Authors state that "covariates for multivariate models were determined on the basis of a review of literature or theoretical assumptions by the co-authors". Meanwhile this may be clear for some covariates, readers may not understand this reasoning for many covariates; for example, experiencing IPV in the past 12 months. Authors should briefly provide the findings and/or thoughts supporting their selection of covariates for the multivariate models other than the sociodemographic characteristics, especially experiencing IPV in the preceding 12 months.

Response: We have now provided additional detail on which co-variables were identified from literature review and which were added based on theoretical assumptions, which has also been justified. The section now reads: "Covariates age, education level, SES-tertile, duration of sex work, having a husband/boyfriend, use of highly effective contraceptives and experience of IPV, were included in the multivariate models on the basis of a review of literature.[8–15,24] Covariates high FP knowledge, positive attitude to FP and high FP specific self-efficacy were included in the multivariate models on the basis of theoretical assumptions by the co-authors. It was hypothesised that these characteristics would be positively associated with contraceptive use and would protect against experiencing an induced abortion."

Ethics

- Even though author indicated having obtained ethical approval, the ethical procedures applied in this study are not written in the text. Ethical considerations should be included in the manuscript.

Response: We have added a paragraph in the methods section entitled 'ethical considerations'. Here we added the following text: "All study participants provided written informed consent. The study was approved by the Kenyatta National Hospital and University of Nairobi Ethics and Research Committee, Kenya (reference KNH-ERC/RR/493) and the Monash University Human Research Ethics Committee, Australia (reference CF16/1552—2016000812)."

Results

- Lines 179-181: I Suggest this section should be moved to the relevant section under "materials and methods".

Response: Transferred this to Materials and methods.

Discussion

- The researchers have attributed the “actual reduction in abortions in this population (line 246)” to unmet needs for induced abortion among FSWs, due to difficulty in accessing abortion services. Could they discuss any ongoing interventions in Kenya/Mombasa targeting FSWs for SRH or any reported programmatic challenges in providing SRH in this population. Could they also discuss any other potential motivations that can trigger an intention to abort or keep a pregnancy in FSWs?

Response: We appreciate this comment, and we would like to clarify that we hypothesized a number of potential explanations for the lower-than-expected rates of induced abortions, including the high rates of highly effective contraceptives in our population, and the ‘lower-than expected HIV prevalence and unintended pregnancy incidence in our cohort and could be a result of peer-mediated interventions implemented over the past years in the Mombasa area’. Also, the fact that this trial attempted to draw a representative sample is mentioned. Indeed, our last-mentioned hypothesized explanation is based on the gap between lifetime unintended pregnancies and lifetime induced abortions could indicate a high unmet need for induced abortion services among FSWs. As requested by the reviewer, we have now added a motivation that FSW may decide to keep the unintended pregnancy, and have provided a reference substantiating this argument. Moreover, we have added a sentence and a reference supporting the statement that there may be challenges in accessing SRH services. The section now reads: ‘The gap between lifetime unintended pregnancies and lifetime induced abortions could indicate that many women decide to keep a child from unintended pregnancies, which could be supported by the fact that between 70-80% of young FSW in Mombasa have reported one or more children.[4,27] It may also indicate a high unmet need for induced abortion services among FSWs, for example due to ongoing or increasing difficulties in accessing SRH or abortion services for this group or increasing sociocultural barriers to abortion. Barriers to accessing other SRH services such as long-acting reversible contraceptives has previously been reported for this population.[28]’

- Line 273-274: Authors state that “...30% of women having their most recent abortions in the informal sector”. In table 1 of the main text, 29 out of 102 FSWs had their most recent induced abortion in the informal sector (pharmacy, traditional healer and home), which corresponds to 28% and not 30% as reported.

Response: This was very well noted by the reviewer, and indeed this was a typographical error, it should have been 29.1% as we had reported in the results section. Please note that all figures in table 1 and table 2 are cluster-adjusted means and proportions, and outcomes of cluster-adjusted bi- and multivariable regression analysis. This explains why the 29/102 does not equate to the 29.1%.

Conclusions

- Line 329-330: I suggest authors should avoid using the phrase; “this was to our knowledge the first study attempting to identify predictors of abortion in FSWs” in the conclusion.

Response: As suggested by the reviewer, we have now deleted the statement from the conclusion.

Supplemental information

- Table 1 & 2: The STROBE checklist would have rather been used to report this study.

Response: We would argue that the actual study design from which the data is derived for this secondary analysis was a cluster-Randomized trial design. We feel it is very important that the design is clearly communicated, as was also communicated by the Editor. We would therefore like to request the reviewer to reconsider this comment, and allow the reporting of the CONSORT in the supplemental information. We have however considered the STROBE checklist and added relevant details to the manuscript where this was important for understanding the cohort analysis. As an

example, we added details on the follow-up time as follows: 'Total follow-up time was 9468 months, with an average of 12.2 months per woman (data not shown).'

VERSION 2 – REVIEW

REVIEWER	Awungafac, George Karolinska Institute, Global Public Health
REVIEW RETURNED	26-Jan-2022

GENERAL COMMENTS	Pages 24-29: The CONSORT checklist is not appropriate for reporting observational studies. You would realize that the responses to the CONSORT checklist items do not apply to the current study/analysis. For example; "Identification as a randomized controlled trial in the title"-which does not apply to this study. Authors are once again advised to fill the STROBE checklist in the current analysis.
---

VERSION 2 – AUTHOR RESPONSE

COMMENT:

Pages 24-29: The CONSORT checklist is not appropriate for reporting observational studies. You would realize that the responses to the CONSORT checklist items do not apply to the current study/analysis. For example; "Identification as a randomized controlled trial in the title"-which does not apply to this study. Authors are once again advised to fill the STROBE checklist in the current analysis.

RESPONSE TO COMMENT:

As suggested by the reviewer, we have now changed the checklist to the STROBE checklist, as attached. In addition, we added the following text (under 'Limitations'): 'The robust multi-stage sampling method improved the ability to generalise the findings to the larger sex work population in the Coast region. However, this study was done in a well-researched population, targeted by other peer-mediated interventions in the past two decades, which may limit generalisation to sex worker populations in other settings.'